# Machine Learning Algorithm for Predicting Distant Metastasis of T1 and T2 Gallbladder Cancer Based on SEER Database

**DOI:** 10.3390/bioengineering11090927

**Published:** 2024-09-15

**Authors:** Zhentian Guo, Zongming Zhang, Limin Liu, Yue Zhao, Zhuo Liu, Chong Zhang, Hui Qi, Jinqiu Feng, Peijie Yao, Haiming Yuan

**Affiliations:** 1Department of General Surgery, Beijing Electric Power Hospital, State Grid Corporation of China, Capital Medical University, Beijing 100073, China; zhentian990305@163.com (Z.G.); liulimin15@sina.com (L.L.); zhaoyue806@sina.com (Y.Z.); lz0002@163.com (Z.L.); zhangchong2118@163.com (C.Z.); qihui6@gt.cn (H.Q.);; 2Key Laboratory of Geriatrics (Hepatobiliary Diseases) of China General Technology Group, Beijing 100073, China; 18811777978@163.com (J.F.); ypj5209@163.com (P.Y.)

**Keywords:** machine learning, SEER, gallbladder cancer, distant metastasis

## Abstract

(1) Background: This study seeks to employ a machine learning (ML) algorithm to forecast the risk of distant metastasis (DM) in patients with T1 and T2 gallbladder cancer (GBC); (2) Methods: Data of patients diagnosed with T1 and T2 GBC was obtained from SEER, encompassing the period from 2004 to 2015, were utilized to apply seven ML algorithms. These algorithms were appraised by the area under the receiver operating characteristic curve (AUC) and other metrics; (3) Results: This study involved 4371 patients in total. Out of these patients, 764 (17.4%) cases progressed to develop DM. Utilizing a logistic regression (LR) model to identify independent risk factors for DM of gallbladder cancer (GBC). A nomogram has been developed to forecast DM in early T-stage gallbladder cancer patients. Through the evaluation of different models using relevant indicators, it was discovered that Random Forest (RF) exhibited the most outstanding predictive performance; (4) Conclusions: RF has demonstrated high accuracy in predicting DM in gallbladder cancer patients, assisting clinical physicians in enhancing the accuracy of diagnosis. This can be particularly valuable for improving patient outcomes and optimizing treatment strategies. We employ the RF algorithm to construct the corresponding web calculator.

## 1. Introduction

Gallbladder cancer (GBC) is characterized by its insidious onset, early metastasis, and poor prognosis [1,2]. The 5-year survival rates for gallbladder cancer in the T3 and T4 stages are 32.4% and 3.5%, respectively [3,4]. Currently, early diagnostic methods with high specificity and sensitivity for GBC remain lacking, and most GBC cases are detected in the middle to late stages [5]. Research indicates that the incidence of lymph node and distant metastasis (DM) in GBC patients ranges from 17.9% to 64.5%, with the liver, lungs, and peritoneum being the most commonly affected organs by metastasis. [6,7,8]. Among gallbladder cancer patients, those with DM have a poorer prognosis compared to those without [7,9]. Distant metastasis is a strongly correlated factor that significantly impacts the prognosis of GBC [10]. Early assessment of distant metastasis risk is vital for timely intervention and enhancing the prognosis of GBC in the T1 and T2 stages. Although, nomogram is currently the most widely utilized clinical tool. At the same time, there is also a continuous development of new technologies. Machine learning (ML) algorithms are increasingly employed to build clinical models due to their practicality, innovation, and accuracy [11]. Machine learning algorithms hold great promise in harnessing complex and massive clinical data for disease diagnosis and outcome prediction. Previous studies have demonstrated that machine learning offers more advantages compared to traditional big data clinical prediction research methods [12].

Therefore, our objective is to utilize an ML algorithm to predict the occurrence of DN in GBC. Our study can assist clinicians in making more precise predictions, thus enhancing their ability to tailor treatment plans to individual patients, improve patient prognosis through early intervention, and enhance the patient prognosis even further.

## 2. Materials and Methods

### 2.1. Data Sources

The patient data originates from the SEER, utilizing SEER*stat 8.4.2 software. This research encompassed individuals who were diagnosed with GBC during the period from 2004 to 2015. The process of patient screening is illustrated in Figure 1. The data criteria are outlined as follows: (1) The 6th edition of the AJCC TNM staging system served as the foundation; (2) Clear histological diagnosis; (3) For a single tumor.

Exclusion criteria: (1) missing or incomplete data, including T staging, M staging, etc. Variables included age, sex, race, tumor size, year of diagnosis, N stage, Hispanic, histology, marital status, T stage, grade, and DM. Distant metastasis refers to the spread of a tumor to distant organs, such as the liver, bone, brain, lung, peritoneum, and other organs. SEER does not require informed consent from the patient concerned, nor does it require ethical approval. This study has received approval from the National Cancer Institute, with reference number 19238-Nov2021.

### 2.2. Model Construction

Statistical analysis of the data in this study was performed utilizing the SPSS software (version 26.0). Construct a nomogram prediction model for DM using R 4.3.2 and draw a calibration curve. All patients were randomly allocated into two subsets (including the training set and the test set) in an 8:2 ratio. Categorical variables are represented in the form of numerical values and corresponding percentages, providing a comprehensive and detailed presentation. Comparisons within the groups are carried out utilizing the chi-square test, Fisher’s exact test, and Mann-Whitney U test. The utilization of these advanced testing methods enhances the depth and sophistication of our research findings, ultimately contributing to a more captivating and illuminating presentation of our results. Additionally, multiple categorical variables are also expressed as numbers and percentages for between-group comparisons, employing the chi-square test and Kruskal-Wallis test. The significance is adjusted using the Bonferroni correction. We establish a logistic regression (LR) based on univariate and multivariate logistic regression analysis and display them as a nomogram. A nomogram is a graphical representation that converts mathematical formulas into geometric expressions and explains the interactions between predicted variables. It is mainly used in logistic regression models and COX proportional risk models [13]. All computed *p*-values were two-sided, and statistical significance was considered at a level of less than 0.05.

Use Python software (version 3.9.12). Include all variables in the machine learning algorithm, and a prediction model is built. In the SEER database, there are fewer cases of DM in T1 and T2 gallbladder cancer, and the original dataset is imbalanced. We use under-sampling and over-sampling techniques to process the raw data. After conducting the sampling, the correlation between variables is illustrated in Figure 2. The darker the color, the higher the negative correlation coefficient; bright colors do the opposite; a and b represent the correlation between data after different sampling methods. We utilized seven commonly used ML algorithms to construct the predictive model, including random forest (RF), decision Tree (DT), support vector machine (SVM), naive Bayes (NB), k nearest neighbor (KNN), eXtreme gradient boosting (XGBoost), and gradient boosting machine (GBM). Model evaluation is mainly based on accuracy, precision, recall, F1 score, and AUC value, and the model with the highest ROC value and F1 score is the optimal model.

## 3. Results

### 3.1. Analysis of Patient Information

Our research encompassed 4371 patients diagnosed with T1 and T2 GBC. Among them, 764 patients presented with distant metastasis, while the remaining 3607 patients did not have distant metastasis. The vast majority of patients were of advanced age (≥70 years old, 56.9%), female (70.3%), and white (76.5%). Significant differences were observed in age, histology, tumor size, grade, N stage, and T stage between patients with or without DM (*p* < 0.05). The information of all patients is detailed in Table 1. During our exploration into the relationship between the training set, test set, and total set, we discovered that there were no significant differences in various indicators among the three datasets, as depicted in Table 2.

We employed LR to identify clinical factors that affect DM. Grade, histology, tumor size, T stage, N stage, and age were all identified as risk factors for DM in gallbladder cancer in univariate and multivariate logistic regression (Table 3). An LR model was developed with an AUC of 0.755 (95% CI: 0.734-0.776) in the test set and an AUC of 0.738 (95% CI: 0.693-0.783) in the training set (Figure 3). Figure 4 displays the calibration curves of LR in different data sets. The calibration curves indicate that the predicted probability curve is closely aligned with the actual values, indicating that LR has good calibration and is consistent with real-world observations. Significant independent factors were incorporated to establish the nomogram (Figure 5a), which clearly illustrates the impact of each risk factor on the outcome. Each factor corresponds to a specific score, which, when combined, provides the predicted probability of DM of early GBC. According to our nomogram, the risk of DM in early GBC ranges from 0.1 to 0.8. The DCA of the DM nomogram (Figure 5b) shows that within the threshold probability range of 1–40%, the net benefit of the model’s decision curve is higher than the net benefit of the two invalid lines. The decision curve illustrates the utility of the model’s predictions for decision-making at different probability thresholds. If the decision curve surpasses the two extreme case curves (i.e., the gray line assumes all patients have the disease, and the black line assumes none), the model is considered valuable.

### 3.2. Analysis of Results Obtained from Machine Learning Algorithms

ML algorithms were evaluated by subsequent metrics, encompassing accuracy, precision, recall, F1 scores, and AUC values. The evaluation of these models was conducted with a comprehensive analysis of their performance across various key indicators. ML algorithms trained using over-sampling data outperformed those trained with under-sampling data. Detailed comparisons of the seven models constructed can be found in Table 4 and Table 5. By employing over-sampling and under-sampling techniques, seven ML algorithms were constructed, and the performance of different data sets is depicted in Figure 6. Among them, the Random Forest (RF) model outperformed the others, achieving an accuracy rate of 0.828, precision of 0.811, recall rate of 0.862, F1 score of 0.836, and an AUC of 0.913. The model also demonstrated excellent performance in the 5-fold cross-validation, with accuracies of 0.808, 0.819, 0.814, 0.817, and 0.812, respectively, with an average accuracy of 0.814, indicating robust performance. The calibration curves of the RF in both the test and training sets are depicted in Figure 7. RF model exhibited strong calibration in different data sets. Feature selection using RF, as shown in Figure 7c, identified grade as a key predictor of DM.

This research has established a web calculator for assessing the risk of DM in T1 and T2 GBC. The tool is intended for use in clinical settings and can be accessed on 1 September 2024 at http://121.43.117.60:8004/.

## 4. Discussion

In this study, we integrated ML algorithms with clinical pathological features to develop a model for DM in gallbladder cancer. Unlike previous studies, this research predicts and analyzes distant metastasis in GBC patients by constructing an ML algorithm. The results, based on the SEER database, demonstrated that among the seven machine learning algorithms compared. Random Forest demonstrated the most superior performance and achieved superior predictive accuracy.

Gallbladder cancer is relatively rare in the population and exhibits a slow increase in incidence. However, within the bile duct system, it is the most prevalent form of malignant tumor. [2,14]. Treatment outcomes are poor once GBC progresses to the middle and late stages. The overall survival (OS) rate for GBC patients is roughly 17.8–21.7%, with a 5-year OS of only 5% [15,16,17]. The 5-year survival rate for T1 stage GBC is between 95% and 100%, whereas, for T3 and T4 stage patients, it drops to only 23% and 12% respectively [18]. The prognosis for GBC patients with distant metastasis is worse than those without, with a 1-year survival rate ranging between 20% and 50% [7,9]. Therefore, assessing the risk of DM in early GBC and developing predictive models are crucial for early identification and clinical intervention, ultimately improving patient prognosis. Currently, research on distant metastasis of gallbladder cancer primarily focuses on disease prognosis, relying mostly on nomograms developed from traditional logistic regression (LR) models or COX competitive risk models [6,19,20]. The traditional logistic regression model evaluates the association between risk factors and specific outcomes, quantifying the strength of the relationship through the generation of corresponding coefficients. However, LR models also have limitations, including sensitivity to multicollinearity and a lack of mechanisms to prevent overfitting [21]. With advancements in artificial intelligence technology, the application of ML in tumor diagnosis and prognosis assessment is becoming increasingly common [22,23]. The ML algorithm also addresses the shortcomings of traditional logistic regression models, such as overfitting and imbalanced data distribution [24]. In this study, we applied the ML algorithm for the first time to predict DM of T1 and T2 stage gallbladder cancer, aiming to improve patient prognosis through early intervention effectively.

This research seeks to apply an ML algorithm to predict DM of T1 and T2 stage gallbladder cancer while identifying the key factors affecting metastasis through logistic regression analysis.

LR analysis identified age, histology, tumor size, T stage, N stage, and grade were all predictive factors for DM in GBC, consistent with previous research findings [6]. Similar to the findings from logistic regression, the feature importance in the RF model also highlights grade as a key predictive variable for evaluating distant metastasis of gallbladder cancer. Tumor grade serves as an indicator to assess the morphological and functional similarity between tumor cells and source organ tissues [25].

Previous studies have also found that grade is a significant predictor of distant metastasis and prognosis in gallbladder cancer patients [6,7,20]. The higher the grade, the poorer the cell differentiation; higher grades are typically associated with greater invasiveness, a broader range of infiltration, and an increased likelihood of DM [20].

Studies have shown that poorly differentiated GBCs are more likely to experience distant metastasis [26], a finding consistent with the conclusion of this study. Lymph node status is a widely recognized predictive factor for evaluating metastasis and prognosis in gastrointestinal malignant tumors [27,28], and a thorough assessment of lymph nodes is also essential for patient treatment [29,30]. This study identified the N stage as an important factor in predicting the occurrence of DM in GBC. This study found that gallbladder cancer patients with tumors 2 cm or larger are more prone to DM, aligning with previous research findings [6].

ML enables computers to emulate human learning capabilities, enhancing performance through iterative data analysis model improvements [31]. Over the past decade, ML algorithms have been extensively applied in the medical field, yielding remarkable results in the diagnosis, treatment, and prognosis of diseases [32]. Compared with traditional data analysis methods, ML offers several key advantages. It can efficiently process large datasets and, more efficiently, handle nonlinear data through diverse algorithms and statistical models, while traditional methods often fall short of achieving satisfactory outcomes with nonlinear data. Numerous studies have demonstrated that the predictive performance of machine learning is superior to traditional methods [13]. In this study, the RF model emerged as one of the most effective ML algorithms. RF employs advanced classification decisions and different weighting ratios, excelling in handling large feature sets and highly nonlinear data. This model enhances the utilization of analytical information, resulting in superior predictive performance [12].

We applied seven predictive models using the SEER to evaluate the distant metastasis in T1 and T2 gallbladder cancer patients. These models were evaluated by accuracy, precision, recall, F1 score, and AUC value. Amongst them, the RF model demonstrated the highest predictive ability (AUC = 0.913, F1 score = 0.836), making it the optimal choice for predicting distant metastasis in gallbladder cancer.

This study also has some limitations: (1) As it relies on North American demographic data, it needs to be validated with external populations in future studies. (2) The performance of the RF algorithm requires further enhancement by incorporating additional risk factors, as well as tumor markers, which may also be important predictive factors for distant cancer metastasis. In response to the above issues, we will gather more comprehensive data and conduct in-depth supplementary studies in future research.

## 5. Conclusions

The efficacy of this model is expected to be further enhanced through the incorporation of additional risk factors, thereby elevating its performance to a higher level and quantitative indicators to forecast DM of T1 and T2 GBC. Among the seven predictive models, the RF algorithm demonstrated superior predictive power. Offer individualized predictive outcomes to enhance patient results. We have applied an online network calculator to assess the risk of DM, which applies to clinical use (http://121.43.117.60:8004/). This tool serves as a valuable resource for evaluating the likelihood of DM in these patient populations.

## Figures and Tables

**Figure 1 bioengineering-11-00927-f001:**
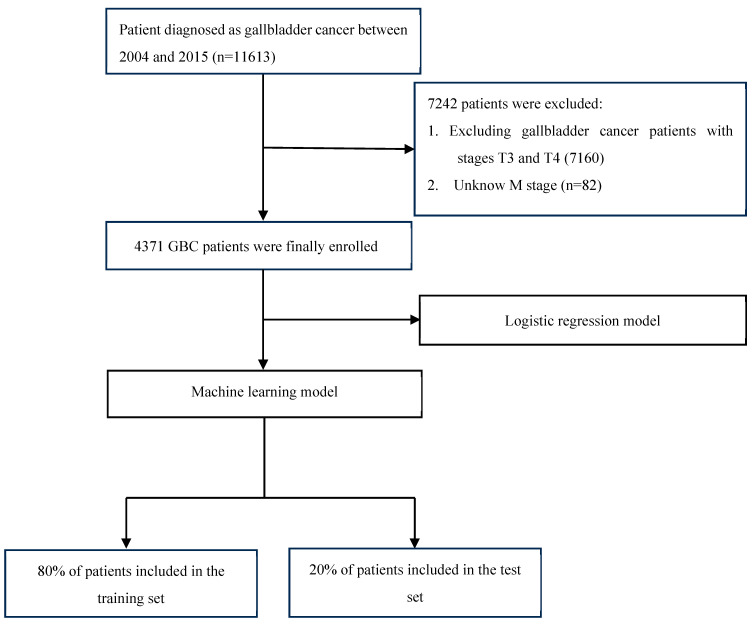
The flow diagram of the selection process for the study.

**Figure 2 bioengineering-11-00927-f002:**
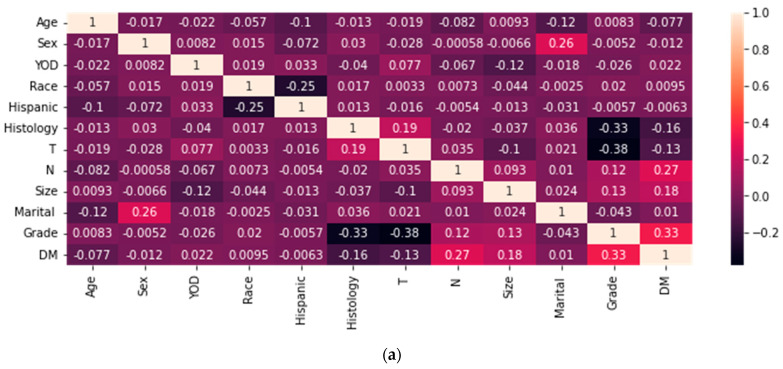
Correlation heatmaps of characteristics are featured in various datasets. (**a**): Data processed using over-sampling. (**b**): Data processed using under-sampling.

**Figure 3 bioengineering-11-00927-f003:**
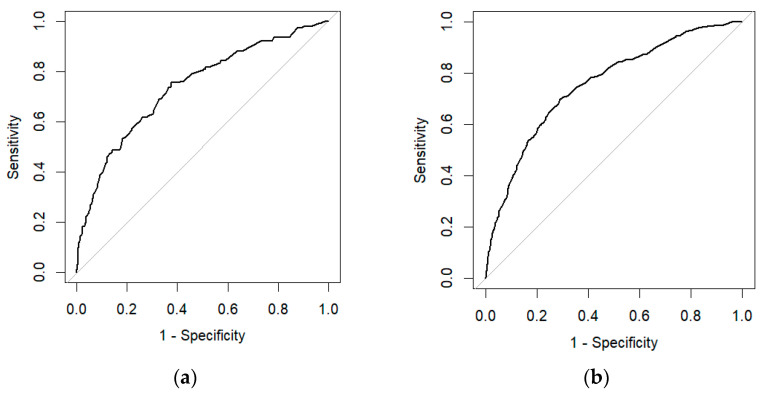
**Prediction of ROC curves for DM in GBC using LR models in the test set and training set.** (**a**): ROC curve generated by the LR model in the test set. (**b**): ROC curve generated by the LR model in the training set.

**Figure 4 bioengineering-11-00927-f004:**
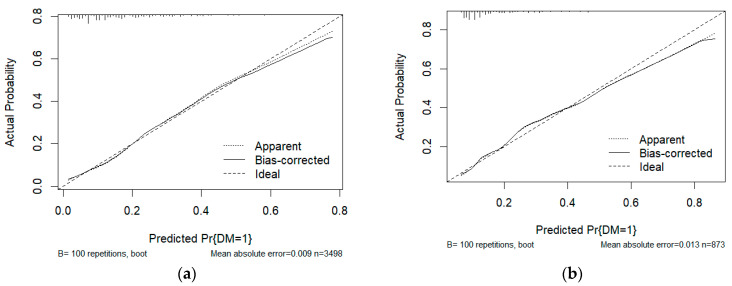
**The calibration plot of the LR.** (**a**): Calibration curve of LR in the test set. (**b**): Calibration curve of LR in the training set.

**Figure 5 bioengineering-11-00927-f005:**
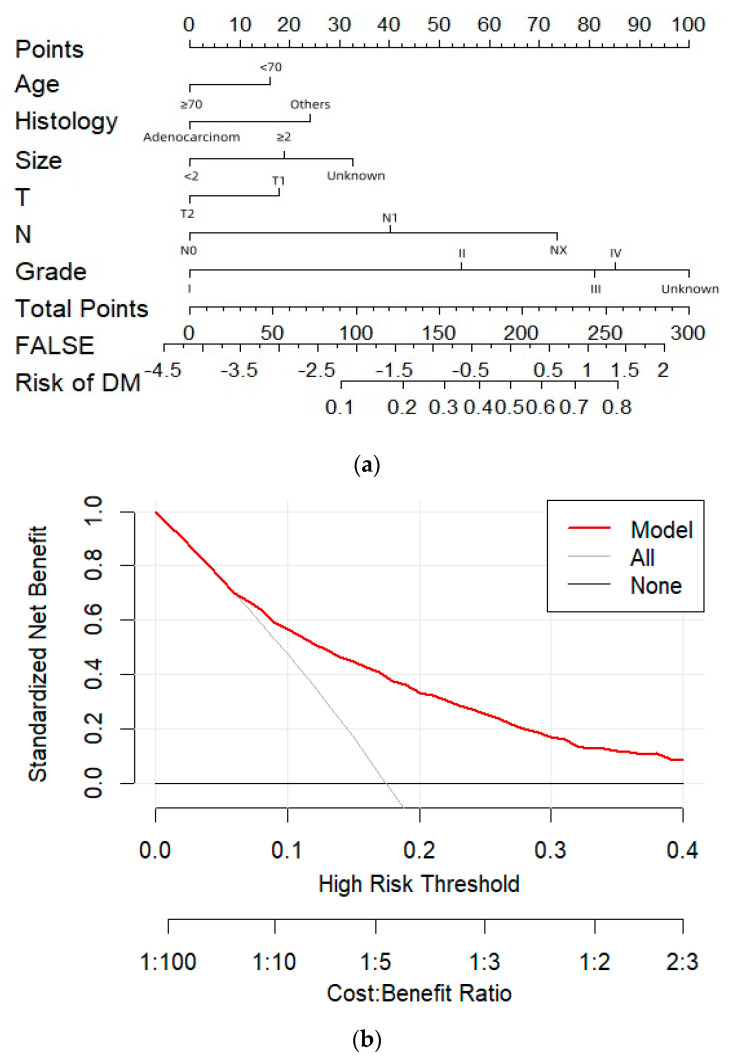
**Nomogram and decision curve for predicting DM of early GBC.** (**a**): The nomogram of the LR. (**b**): Decision curve analysis of GBC distant metastasis.

**Figure 6 bioengineering-11-00927-f006:**
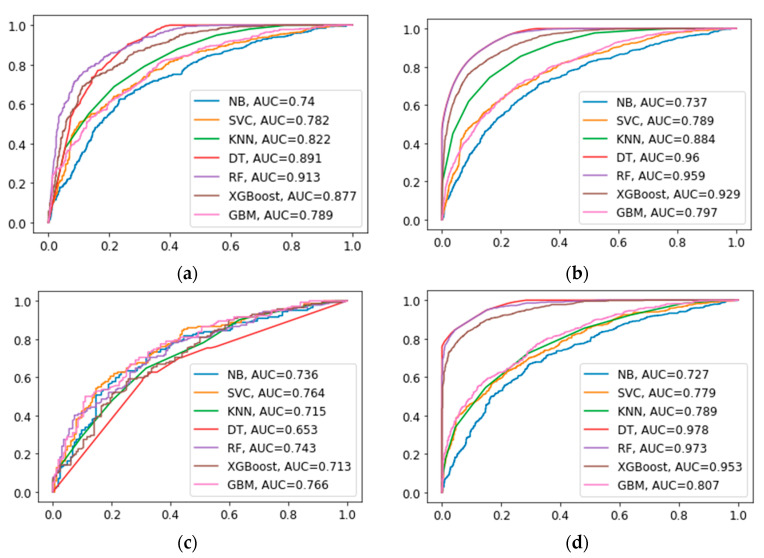
ROC curves for 7 machine learning algorithms across various datasets. (**a**): The ROC curves for the 7 machine learning algorithms in the test set were generated using over-sampling. (**b**): The ROC curves for the 7 machine learning algorithms in the training set were generated using over-sampling. (**c**): The ROC curves for the 7 machine learning algorithms in the test set were generated using under-sampling. (**d**): The ROC curves for the 7 machine learning algorithms in the training set were generated using over-sampling.

**Figure 7 bioengineering-11-00927-f007:**
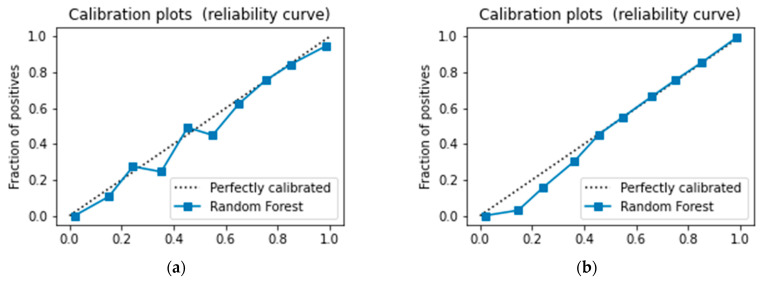
**Calibration plots of RF in training and test sets and the importance of RF features.** (**a**): Calibration curve of RF in the test set. (**b**): Calibration curve of RF in the training set. (**c**): Feature importance derived from the RF.

**Table 1 bioengineering-11-00927-t001:** Characteristics of the gallbladder cancer patients in T1 and T2.

Characteristic	Without DM (N = 3607)	With DM (N = 764)	*p*-Value
Age (Year)			<0.001
<70	1508 (41.8%)	374 (49.0%)	
≥70	2099 (58.2%)	390 (51.0%)	
Gender			0.181
Female	2523 (69.9%)	553 (72.4%)	
Male	1084 (30.1%)	211 (27.6%)	
Race			0.599
White	2770 (76.8%)	578 (75.7%)	
Black	400 (11.1%)	97 (12.7%)	
Other	437 (12.1%)	89 (11.6%)	
Hispanic			0.572
YES	808 (22.4%)	164 (21.5%)	
NO	2799 (77.6)	600 (78.5%)	
Histology			<0.001
Adenocarcinoma	3308 (91.7%)	611 (80.0%)	
Others	299 (8.3%)	153 (20.0%)	
Year of Diagnosis			0.262
2004–2009	1624 (45.0%)	327 (42.8%)	
2010–2015	1983 (55.0%)	437 (57.2%)	
Tumor Size(cm)			<0.001
<2	838 (23.2%)	81 (10.6%)	
≥2	1543 (42.8%)	321 (42%)	
Unknown	1226 (34%)	362(47.4%)	
T Stage			<0.001
T1	1259 (34.9%)	361 (47.3%)	
T2	2348 (65.1%)	403 (52.7%)	
N stage			<0.001
N0	2871 (79.6%)	422 (55.2%)	
N1	644 (17.8%)	257 (33.7%)	
NX	92 (2.6%)	85 (11.1%)	
Marital Status			0.531
Single	1839 (51.0%)	380 (49.7%)	
Married	1768 (49.0%)	384 (50.3%)	
Grade			<0.001.
Grade I	737 (20.4%)	39 (5.1%)	
Grade II	1536 (42.6%)	219 (28.6%)	
Grade III	894 (24.8%)	255 (33.4%)	
Grade IV	55 (1.5%)	18 (2.4%)	
Unknown	385 (10.7)	233 (30.5%)	

**Table 2 bioengineering-11-00927-t002:** The relationship between the training set, testing set, and total set.

Characteristic	Training Set (N = 3498)	Test Set (N = 873)	Total Set (N = 4371)	*p*-Value
Age (Year)				1.000
<70	1506 (43.1%)	376 (43.1%)	1882 (43.1%)	
≥70	1992 (56.9%)	497 (56.9%)	2489 (56.9%)	
Gender				0.386
Female	2445 (69.9%)	631 (72.3%)	3076 (70.4%)	
Male	1053 (30.1%)	242 (27.7%)	1295 (29.6%)	
Race				0.883
White	2686 (76.8%)	662 (75.8%)	3348 (76.6%)	
Black	390 (11.1%)	107 (12.3%)	497 (1134%)	
Other	422 (12.1%)	104 (11.9%)	526 (12.0%)	
Hispanic				0.256
YES	796 (22.8%)	176 (20.2%)	972 (22.2%)	
NO	2702 (77.2)	697 (79.8%)	3399 (77.8%)	
Histology				0.207
Adenocarcinoma	3122 (89.3%)	797 (91.3%)	3919 (89.7%)	
Others	376 (10.7%)	76 (8.7%)	452 (10.3%)	
Year of Diagnosis				0.844
2004–2009	1569 (44.9%)	382 (43.8%)	1951 (44.6%)	
2010–2015	1929 (55.1%)	491 (56.2%)	2420 (55.4%)	
Tumor size(cm)				0.969
<2	735 (21%)	184 (21.1%)	919 (21.1%)	
≥2	1488(42.5%)	376 (43%)	1864 (42.6%)	
Unknown	1275(36.4%)	313 (35.9%)	1588 (36.3%)	
T Stage				0.756
T1	1306 (37.3%)	314 (36%)	1620 (37.1%)	
T2	2192 (62.7%)	559 (64%)	2751 (62.9%)	
N stage				0.104
N0	2611 (74.6%)	682 (78.1%)	3293 (75.3%)	
N1	741 (21.2%)	160 (18.3%)	901 (20.6%)	
NX	146 (4.2%)	31 (3.6%)	177 (4.1%)	
Marital Status				0.998
Single	1775 (50.7%)	444 (50.9%)	2219 (50.8%)	
Married	1723 (49.3%)	429 (49.1%)	2152 (49.2%)	
Grade				0.620
Grade I	619 (17.7%)	157 (18.0%)	776 (17.7%)	
Grade II	1384 (39.6%)	371 (42.5%)	1755 (40.2%)	
Grade III	949 (27.1%)	200 (22.9%)	1149 (26.3%)	
Grade IV	51 (1.5%)	22 (2.5%)	73 (1.7%)	
Unknown	495 (14.1%)	123 (14.1%)	618 (14.1%)	
Distant Metastasis				0.998
YES	612 (17.5%)	152 (17.4%)	764 (17.5%)	
NO	2886 (82.5%)	721 (82.6%)	3607 (82.5%)	

**Table 3 bioengineering-11-00927-t003:** LR analysis in the training cohort.

	Univariable Analysis			Multivariable Analysis		
	OR	95%CI	*p* Value	OR	95%CI	*p* Value
Age (Year)						
<70	Ref			Ref		
≥70	0.723	0.607–0.861	<0.001	0.705	0.583–0.852	<0.001
Gender						
Female	Ref					
Male	0.881	0.726–1.069	0.200			
Race						
White	Ref					
Black	1.116	0.850–1.464	0.431			
Other	0.980	0.746–1.287	0.885			
Hispanic						
YES	0.997	0.810–1.228	0.977			
NO	Ref					
Histology						
Adenocarcinoma	0.345	0.274–0.436	<0.001	0.595	0.456–0.777	<0.001
Others	Ref			Ref		
Year of Diagnosis						
2004–2009	Ref					
2010–2015	1.151	0.965–1.374	0.117			
Tumor Size(cm)						
<2	Ref			Ref		
≥2	1.916	1.449–2.534	<0.001	1.507	1.121–2.027	0.007
Unknown	2.729	2.067–3.602	<0.001	2.023	1.509–2.714	<0.001
T Stage						
T1	Ref					
T2	0.594	0.498–0.708	<0.001	0.679	0.547–0.843	<0.001
N Stage						
N0	Ref			Ref		
N1	2.656	2.155–3.197	<0.000	2.377	1.920–2.944	<0.001
NX	6.067	4.299–8.563	<0.000	4.913	3.398–7.105	<0.001
Marital Status						
Single	Ref					
Married	1.096	0.920–1.305	0.304			
Grade						
Grade I	Ref			Ref		
Grade II	3.507	2.281–5.391	<0.001	3.236	2.090–5.010	<0.001
Grade III	6.835	4.453–10.489	<0.001	5.776	3.721–8.966	<0.001
Grade IV	8.990	4.316–18.725	<0.001	6.316	2.932–13.605	<0.001
Unknown	13.936	8.977–21.635	<0.001	8.684	5.475–13.774	<0.001

**Table 4 bioengineering-11-00927-t004:** Different model performance in over-sampling.

Model	Accuracy	AUC	Precision	Recall Rate	F1-Score
NB	0.681	0.740	0.734	0.587	0.652
SVC	0.707	0.782	0.722	0.690	0.706
KNN	0.738	0.822	0.721	0.791	0.761
DT	0.681	0.891	0.686	0.688	0.687
RF	0.828	0.913	0.811	0.862	0.836
XGBoost	0.784	0.877	0.781	0.799	0.790
GBM	0.704	0.789	0.711	0.704	0.707

**Table 5 bioengineering-11-00927-t005:** Different model performance in under-sampling.

Model	Accuracy	AUC	Precision	Recall Rate	F1-Score
NB	0.689	0.736	0.715	0.549	0.621
SVC	0.702	0.764	0.691	0.647	0.669
KNN	0.604	0.715	0.562	0.661	0.687
DT	0.699	0.653	0.676	0.676	0.676
RF	0.686	0.743	0.643	0.725	0.682
XGBoost	0.656	0.713	0.624	0.654	0.639
GBM	0.702	0.766	0.683	0.669	0.676

## Data Availability

Details of the data access process are available online. This data can be accessed on 1 September 2024. (https://github.com/990305/GBC).

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
