# Peer review of "Machine Learning Algorithm for Predicting Distant Metastasis of T1 and T2 Gallbladder Cancer Based on SEER Database"

_bioengineering, 2024, doi:10.3390/bioengineering11090927_

Round 1
Reviewer 1 Report
Comments and Suggestions for Authors
The authors develop a machine learning (ML) algorithm to predict the risk of distant metastasis (DM) in T1 and T2 gallbladder cancer (GBC) patients. They used demographic and clinical data from the NIH’s SEER database (2004-2015) to create seven ML models. The work seems interesting. However, I have the following comments.
- The authors state in the abstract " we construct a machine learning algorithms". I think it must be apply instead of construct. constructing an algorithm means the authors created it.
- In figure 3, I suggest that the authors starting the x-axis from 1 or to start from 0 and relabel the acis 1-specificity.
- The authors mentioned "preventing overfitting" but how can they check. They need to evaluate the training vs testing performance to validate over/underfitting.
-Figures 6-7 are fuzzy, please provide a better quality versions of the figures.
Author Response
- Comment: the authors state in the abstract " we construct a machine learning algorithms". I think it must be apply instead of construct. constructing an algorithm means the authors created it.
Response: We have revised this statement according to your suggestion to: "We aim to apply a machine learning (ML) algorithm to predict the risk of distant metastasis (DM) of T1 and T2gallbladder cancer (GBC) ". Page 1, Line 15-16.
- Comment: In figure 3, I suggest that the authors starting the x-axis from 1 or to start from 0 and relabel the acis 1-specificity.
Response: We have made modifications to the graphics according to your suggestion, as shown in Figure 3. (Page 7, Line 156).
- Comment: The authors mentioned "preventing overfitting" but how can they check. They need to evaluate the training vs testing performance to validate over/underfitting.
Response: Over-fitting is usually better than models that are too complex, resulting in better performance in the training set and poor performance in the testing set. The optimal model RF in this paper demonstrated excellent predictive performance in both the training and testing sets, with an AUC of 0.958 for the training set; The AUC of the test set was 0.913, and we used 5-fold crossvalidation to test the model. The accuracy were 0.808, 0.819, 0.814, 0.817, and 0.812, respectively, with an average accuracy of 0.814(Page 9, Line 177), indicating that the model performed well; Under-fitting is superior to simpler models, resulting in poor performance in both the training and testing sets, while RF models perform well in both sets.
- Comment: Figures 6-7 are fuzzy, please provide a better quality versions of the figures.
Response: We have made changes to Figure 7 to facilitate readers' better reading of the article. Regarding Figure 6, considering the abundance of images and the relationship between curves and fonts, we believe that the current form may be more conducive to reading. Thank you again for your question.

Reviewer 2 Report
Comments and Suggestions for Authors
Guo et al leveraged a large public database to predict the occurrence of distant metastasis in gallbladder cancer and associated risk factors. They collected > 10,000 patients from the SEER database and excluded those didn’t match the inclusion criterion. The dataset was split into training and testing cohort to benchmark 7 classification models. Random forest was claimed as the best model (Though I’m not sure how this was selected since RF was not among the top in several metrics). The overall design is straightforward. However, there are some issues that detract enthusiasm for this report. My major concern is that it’s not clear to me what’s the main goal of the paper. These 7 models have been used in different biological settings, and we know their performance largely depends on the data and the problem you are trying to solve. What’s novel here? What are the factors significantly related to the outcomes and how would they guide the treatment? When training with different data, the model will change, how could clinicians apply this?
Other comments
The patients collected till 2015, not 2021.
LR was not defined throughout the manuscript
How were factors screened by uni-/multi- variable analysis?
When fitting the models with tons of variables, did the author consider their collinearity as this may significantly affect the performance and stability.
P-value adjusted for multiple test?
Figures are in low quality. Eg. Figure2, 7 etc. Fonts are too small to read
I recommend the authors carefully check any typos/errors/syntax across the manuscript (There are multiple).
Comments on the Quality of English LanguageI recommend the authors carefully check any typos/errors/syntax across the manuscript (There are multiple).
Author Response
- Comment: Random forest was claimed as the best model (Though I’m not sure how this was selected since RF was not among the top in several metrics.
Response: In this study, the RF model established using oversampling performed better than the other six models in the test set, with evaluation metrics including accuracy of 0.828, precision 0.811, recall rate 0.862, F1 score 0.836, and AUC 0.913, From Table 3, it can be seen that all five evaluation indicators are superior to the other six models. (Page 9, Line 191-192).
- Comment: The overall design is straightforward. However, there are some issues that detract enthusiasm for this report. My major concern is that it’s not clear to me what’s the main goal of the paper.
Response: Our research aims to establish a machine learning model that can predict distant metastasis in T1 and T2 stage gallbladder cancer patients. This study can provide clinicians with more personalized clinical decisions, improve patient prognosis through early intervention, and effectively enhance patient quality of life.At the same time, this study also used traditional logistic regression models to screen for risk factors for TI and T2 stage metastasis of gallbladder cancer. Although its predictive performance is not as good as machine learning algorithms, it still has certain advantages, such as interpretability. Risk factors can also prompt doctors to pay more attention to patients who are prone to distant metastasis.
- Comment: These 7 models have been used in different biological settings, and we know their performance largely depends on the data and the problem you are trying to solve. What’s novel here? What are the factors significantly related to the outcomes and how would they guide the treatment? When training with different data, the model will change, how could clinicians apply this?
Response: Firstly, as you mentioned, machine learning algorithms have been widely applied in various biological environments,machine learning algorithms have broad prospects in utilizing complex and massive clinical data for disease diagnosis and outcome prediction. Previous studies have shown that machine learning has more advantages than traditional big data clinical prediction research methods. Secondly, there is currently no scholar using machine learning algorithms to predict distant metastasis of gallbladder cancer in this research question. In previous studies, the performance of the traditional regression model established was relatively average, with an AUC of only 0.679 in the test set. Although it can provide some advice for clinical doctors, its credibility needs to be improved[1]. In terms of predictive performance, the RF model established in our paper outperforms traditional logistic regression models (AUC=0.913 in the RF test set, while AUC=0.755 in the traditional logistic regression test set), and the RF model has also achieved good results in other evaluation metrics such as accuracy and precision.Gallbladder cancer, as a common malignant tumor of the biliary tract, is prone to early distant metastasis[2], the 5-year survival rate for stage IV patients is only 1%[3]. For T1 and T2 stage gallbladder cancer patients with distant metastasis, combination therapy including surgery and chemotherapy may be crucial for prolonging patient survival[4, 5]. Therefore, it is urgent to establish an accurate and efficient predictive model to help clinical doctors screen patients with early gallbladder cancer who have distant metastasis. We have already established a webpage (http://121.43.117.60:8004/) based on the model established by the best algorithm for the convenience of clinical doctors.
- Comment: The patients collected till 2015, not 2021.
Response: This is due to the limitations of the SEER database itself. In order to include as many cases as possible, we chose the AJCC6 staging criteria and collected patients from 2004 to 2015, while AJCC7 can only collect patients from 2010 to 2015, and AJCC8 collects patient data from around 2018 to 2021. At the same time, the collection of other data also limits the scope of our data collection. For patients beyond 2015, data such as tumor size and differentiation degree in the SEER database are missing. Therefore, we chose data from 2004 to 2015, At this point, At this point, other studies have also made similar or identical choices[1, 6, 7].
- Comment: LR was not defined throughout the manuscript.
Response: According to your feedback, we have made modifications. (Page1 , Line 21-22; Page2, Line 83.
- Comment: How were factors screened by uni-/multi- variable analysis?
Response: We first conduct a preliminary selection of factors that have an impact on the outcome through univariate analysis, and then conduct a multivariate analysis based on the univariate analysis to analyze all possible influencing factors at once. The results of univariate analysis are often affected by confounding factors and are not very reliable; Multi factor analysis can correct the influence of various confounding factors, and the results are more reliable.
- Comment: When fitting the models with tons of variables, did the author consider their collinearity as this may significantly affect the performance and stability.
Response: Firstly, machine learning algorithms do not need to consider collinearity to some extent. The biggest problem with collinearity is that it affects the interpretability of the parameters of the linear model (i.e. leading to unstable model parameter estimation and increased standard error). From a goal perspective, machine learning is not concerned about this issue and chooses to sacrifice interpretability in pursuit of accuracy. Machine learning models, such as neural networks, support vector machines, etc., have strong nonlinear fitting capabilities and can automatically learn the correlation and weight allocation between input features. This ability enables machine learning models to adjust parameters through learning when facing collinearity problems, in pursuit of higher accuracy and generalization ability, without much concern for parameter interpretability. In extreme cases, excessive collinearity may affect the interpretability of the model, causing overfitting and even affecting the stability and convergence speed of the training process. For example, when there is significant collinearity among multiple variables, it may affect the generalization error of the model and reduce its predictive ability. However, for most practical applications, machine learning models can handle collinearity problems well, as long as the accuracy and generalization ability of the model are guaranteed, collinearity problems will not become the main obstacle. From the results, our RF machine learning model has excellent accuracy and predictive performance, as well as performance on both the training and testing sets. Therefore, the collinearity problem has a relatively small impact on our machine learning model.
For the multivariate logistic regression model, we used the commonly used indicator for determining collinearity, which is the Variance Inflation Factor (VIF) value. The reason why it is called variance inflation factor is because VIF measures the correlation between the variance of parameter estimates and other independent variables. The larger the VIF value, the more severe the multicollinearity. It is generally believed that when VIF is greater than 10 (strictly 5), the model has serious collinearity problems. In this logistic regression model, we used VIF to determine multicollinearity. The results showed that all six variables that make up the multivariate regression model are less than 2, so we did not consider the impact of multicollinearity on the model.
- Comment: P-value adjusted for multiple test?
Response: The reason why binary logistic regression can use multiple tests to adjust p-values is to control false positive rates and ensure the accuracy of statistical inference. If multiple tests are not adjusted, it may lead to an increase in false positive rates due to the cumulative effect of multiple tests. This type of error is referred to as the first type of error in statistics, also known as false positive error. The SPSS software we use does not directly provide multiple tests to adjust the P-value, but we used methods such as Walds test, likelihood ratio test, and fit test to control the probability of type 1 errors and ensure the reliability and validity of the research results.
- Comment: Figures are in low quality. Eg. Figure2, 7 etc. Fonts are too small to read
Response: We have made revisions to the article based on your suggestions.
- Comment: I recommend the authors carefully check any typos/errors/syntax across the manuscript (There are multiple).
Response: We have made revisions to the article based on your suggestions. We have sought English experts to thoroughly revise the article.
References
[1] Cai YL, Lin YX, Jiang LS, Ye H, Li FY, Cheng NS. A Novel Nomogram Predicting Distant Metastasis in T1 and T2 Gallbladder Cancer: A SEER-based Study. Int J Med Sci, 2020, 17(12): 1704-1712. doi: 10.7150/ijms.47073.
[2] Fong Y, Jarnagin W, Blumgart LH. Gallbladder cancer: comparison of patients presenting initially for definitive operation with those presenting after prior noncurative intervention. Ann Surg, 2000, 232(4): 557-69. doi: 10.1097/00000658-200010000-00011.
[3] Donohue JH, Stewart AK, Menck HR. The National Cancer Data Base report on carcinoma of the gallbladder, 1989-1995. Cancer, 1998, 83(12): 2618-28. doi: 10.1002/(sici)1097-0142(19981215)83:12<2618::aid-cncr29>3.0.co;2-h.
[4] Primrose JN, Fox RP, Palmer DH, Malik HZ, Prasad R, Mirza D, Anthony A, Corrie P, Falk S, Finch-Jones M, Wasan H, Ross P, Wall L, Wadsley J, Evans JTR, Stocken D, Praseedom R, Ma YT, Davidson B, Neoptolemos JP, Iveson T, Raftery J, Zhu S, Cunningham D, Garden OJ, Stubbs C, Valle JW, Bridgewater J. Capecitabine compared with observation in resected biliary tract cancer (BILCAP): a randomised, controlled, multicentre, phase 3 study. Lancet Oncol, 2019, 20(5): 663-673. doi: 10.1016/s1470-2045(18)30915-x.
[5] Ben-Josef E, Guthrie KA, El-Khoueiry AB, Corless CL, Zalupski MM, Lowy AM, Thomas CR, Jr., Alberts SR, Dawson LA, Micetich KC, Thomas MB, Siegel AB, Blanke CD. SWOG S0809: A Phase II Intergroup Trial of Adjuvant Capecitabine and Gemcitabine Followed by Radiotherapy and Concurrent Capecitabine in Extrahepatic Cholangiocarcinoma and Gallbladder Carcinoma. J Clin Oncol, 2015, 33(24): 2617-22. doi: 10.1200/jco.2014.60.2219.
[6] Mao W, Deng F, Wang D, Gao L, Shi X. Treatment of advanced gallbladder cancer: A SEER-based study. Cancer Med, 2020, 9(1): 141-150. doi: 10.1002/cam4.2679.
[7] Jiang Y, Jiang L, Li H, Yuan S, Huang S, Fu Y, Li S, Li F, Li Q, Yan X, Chen J, Liu J. Adjuvant chemoradiotherapy in resected gallbladder cancer: A SEER-based study. Heliyon, 2023, 9(3): e14574. doi: 10.1016/j.heliyon.2023.e14574.

Reviewer 3 Report
Comments and Suggestions for Authors
The Authors applied a classical method of logistic regression to "predict the risk of distant metastasis (DM) of T1 and T2 gallbladder cancer (GBC)". The learning stage (i.e. building a model via a machine learning algorithm) was preceded by data analysis (mostly statistical) to establish the most important features, influencing the output. Such an approach was presented in thousands of papers, which described the process of learning from data. I can't see any original contribution of the Authors, which would distinguish their approach from known in the literature.
I rated low the quality of the presentation. In fact, the results presented graphically are poorly commented/discussed (or even not commented) in the text, e.g. Figure 2, Figure 5, etc. Also the data characteristics presented in Tables 1 and 2 does not allow more precise information about the complexity of the problem, e.g. data overlapping, what limitates the possibilities of separating data clusters.
The Authors present the prediction results for 7 models (e.g. in Figure 6). However no detailed information of those models are provided, what makes impossible to use the results presented in the paper to solve the own problems of the readers.
There are also minor questions:
-- why the ranged of the axes to present the ROC curves in Fig. 3 are a little bit strange, while the axes in Fig. 6 are "traditionally correct"?
-- what do the terms "over-sampling" and "under-sampling" mean precisely?
Comments on the Quality of English LanguageThe text should be corrected by a native speaker, because some incorrects grammar constructions make troubles to understand the content. For example the sentence (lines 93-95): "In the SEER database, there are fewer cases of distant metastasis in T1 and T2 gallbladder cancer patients, the original dataset is imbalanced.".
Also there is a permanent editorial error in the text - there must be spaces after punctuation marks (',', '.', ':', etc.).
Author Response
- Comment: The Authors applied a classical method of logistic regression to "predict the risk of distant metastasis (DM) of T1 and T2 gallbladder cancer (GBC)". The learning stage (i.e. building a model via a machine learning algorithm) was preceded by data analysis (mostly statistical) to establish the most important features, influencing the output. Such an approach was presented in thousands of papers, which described the process of learning from data. I can't see any original contribution of the Authors, which would distinguish their approach from known in the literature.
Response: First of all, as you mentioned, using machine learning and traditional logistic regression to solve practical problems is a common paper routine. However, the core of our paper is to use existing tools to solve practical clinical problems and improve patient prognosis. According to our literature search, we first applied machine learning algorithms to the study of predicting distant metastasis of early gallbladder cancer, and found that the predictive performance of machine learning algorithms is superior to traditional logistic regression, which can provide more accurate references for clinical doctors. However, traditional logistic regression also has its advantages, such as better interpretability compared to machine learning. By identifying risk factors, it can have a certain warning effect on clinical doctors. Gallbladder cancer, as a common malignant tumor of the biliary tract, is prone to early distant metastasis[2], the 5-year survival rate for stage IV patients is only 1%[3]. For T1 and T2 stage gallbladder cancer patients with distant metastasis, combination therapy including surgery and chemotherapy may be crucial for prolonging patient survival[4,5]. Therefore, it is urgent to establish an accurate and efficient predictive model to help clinical doctors screen patients with early gallbladder cancer who have distant metastasis. We have already established a webpage (http://121.43.117.60:8004/) based on the model established by the best algorithm for the convenience of clinical doctors.
- Comment: I rated low the quality of the presentation. In fact, the results presented graphically are poorly commented/discussed (or even not commented) in the text, e.g. Figure 2, Figure 5, etc. Also the data characteristics presented in Tables 1 and 2 does not allow more precise information about the complexity of the problem, e.g. data overlapping, what limitates the possibilities of separating data clusters.
Response: Following your suggestion, we have provided explanations for Figure 2(Page 3, Line 97-100) and Figure 5(Page 5, Line 137-150).
Secondly, regarding your issue of limited information improvement in Tables 1 and 2, we have provided the original data https://github.com/990305/GBC. Moreover, Tables 1 and 2 are also common data analysis formats for clinical research papers[8, 9].
- Comment: The Authors present the prediction results for 7 models (e.g. in Figure 6). However no detailed information of those models are provided, what makes impossible to use the results presented in the paper to solve the own problems of the readers.
Response: Your question is very crucial. For ordinary clinical doctors, it is very difficult to use R or Python to make machine learning models, which is not conducive to the promotion of the model. However, machine learning has great advantages and accuracy in predicting distant metastasis of early gallbladder cancer due to its powerful performance. It can provide practical help to clinical doctors. Therefore, we have used a large amount of data from SEER to train the best model and established a webpage. Simply fill in the indicator data to obtain the prediction probability.
- Comment: why the ranged of the axes to present the ROC curves in Fig. 3 are a little bit strange, while the axes in Fig. 6 are "traditionally correct"?
Response: We have made modifications to Figure 3. (Page 7, Line 153)
- Comment: what do the terms "over-sampling" and "under-sampling" mean precisely?
Response: Over-sampling and under-sampling in machine learning are two common data processing techniques used to address imbalanced datasets. Over-sampling refers to increasing the number of minority class samples to match the number of majority class samples. This can help the model better learn the features of minority classes and improve the classifier's predictive performance for minority classes. The methods of oversampling include copying samples, generating synthetic samples, etc. We use the SMOTE method to generate synthetic samples, and the basic idea of SMOTE is to generate new synthetic samples by interpolating between minority class samples. Under-sampling refers to reducing the number of majority class samples to match the number of minority class samples. This can reduce the impact of majority class samples on the model and improve the classification performance for minority classes. The under-sampling methods include random deletion of samples, clustering methods, etc. Using a balanced dataset to train machine learning models can better handle classification problems for positive examples (minority classes).
- Comment: The text should be corrected by a native speaker, because some incorrects grammar constructions make troubles to understand the content. For example the sentence (lines 93-95): "In the SEER database, there are fewer cases of distant metastasis in T1 and T2 gallbladder cancer patients, the original dataset is imbalanced.".Also, there is a permanent editorial error in the text - there must be spaces after punctuation marks (',', '.', ':', etc.).
Response: We are very sorry for the negative impression caused to you on this issue. We have carefully corrected the manuscript under the guide of English professor in the revised version to meet the publishing requirements.
References
[1] Chen Q, He L, Li Y, Zuo C, Li M, Wu X, Pu C, Xu X, Tang R, Xiong Y, Li J. Risk Factors on the Incidence and Prognostic Effects of Colorectal Cancer With Brain Metastasis: A SEER-Based Study. Front Oncol, 2022, 12: 758681. doi: 10.3389/fonc.2022.758681.
[2] Ren B, Yang Y, Lv Y, Liu K. Survival outcome and prognostic factors for early-onset and late-onset metastatic colorectal cancer: a population based study from SEER database. Sci Rep, 2024, 14(1): 4377. doi: 10.1038/s41598-024-54972-3.

Round 2
Reviewer 2 Report
Comments and Suggestions for Authors
The authors have replied to most of the comments. However, it's still unclear about the statistical test you used for the multi-group (>2) test, and how you adjusted the P values. As I know, the SPSS does provide post-hoc analysis.
Author Response
Comment 1: The authors have replied to most of the comments. However, it's still unclear about the statistical test you used for the multi-group (>2) test, and how you adjusted the P values. As I know, the SPSS does provide post-hoc analysis.
Response 1: Thank you for your inquiry. In response to your request, we have reanalyzed the data in the paper and divided it into a training set, test set, and total set. This approach not only provides clarity on our data but also enhances readers' understanding. For multi-group analysis, we utilized the chi-square test and Kruskal-Wallis test, adjusting the significance P-values using the Bonferroni correction method. We have provided an explanation in the methodology section (Page 7, Line 82-87). The data analysis is detailed in Table 2. When using Kruskal-Wallis test, multiple comparisons (Bonferroni) were not performed as the overall test did not detect significant differences between samples.